# Selective atomic sieving across metal/oxide interface for super-oxidation resistance

Shuang Li [1,6], Li Yang[2,6], Jijo Christudasjustus [3], Nicole R. Overman[4], Brian D. Wirth[2], Maria L. Sushko [3], Pauline Simonnin [3], Daniel K. Schreiber [4] ✉, Fei Gao [5] ✉ & Chongmin Wang [1] ✉

Surface passivation, a desirable natural consequence during initial oxidation of alloys, is the foundation for functioning of corrosion and oxidation resistant alloys ranging from industrial stainless steel to kitchen utensils. This initial oxidation has been long perceived to vary with crystal facet, however, the underlying mechanism remains elusive. Here, using in situ environmental transmission electron microscopy, we gain atomic details on crystal facet dependent initial oxidation behavior in a model Ni-5Cr alloy. We find the (001) surface shows higher initial oxidation resistance as compared to the (111) surface. We reveal the crystal facet dependent oxidation is related to an interfacial atomic sieving effect, wherein the oxide/metal interface selectively promotes diffusion of certain atomic species. Density functional theory calculations rationalize the oxygen diffusion across Ni(111)/NiO(111) interface, as contrasted with Ni(001)/NiO(111), is enhanced. We unveil that crystal facet with initial fast oxidation rate could conversely switch to a slow steady state oxidation.

Self-passivation is a fundamental operating principle for corrosion and oxidation resistance of metals and alloys, by which a passivating layer, oftentimes several nanometers in thickness, naturally forms even at room temperature to inhibit further oxidation[1-12]. Different surface facet of alloy exhibits a different atomic arrangement, coordination, and an associated potential energy landscape that subsequently leads to unique electronic, physical, and/or chemical properties[4,5,13-19]. Consequently, one of the critical questions in terms of metal and alloy passivation is how surface crystallographic structure affects the oxidation or corrosion behavior during both initial and subsequently persistent, long-term oxidation.

Indeed, many studies have documented variability in the oxidation rate on different crystal facets of Ni-based alloy[20-22] with diverging conclusions. For the case of initial oxidation of Ni and Ni-Cr alloy, both oxide nucleation and growth rate show surface crystallographic structure dependence[23-26]. During long-term oxidation of Ni and Ni-Cr alloys, a general consensus is that the oxidation rate on the (001)

surface is faster than that on the (111) surface[27,28]. Conversely, a faster oxidation rate on (111) than on (001) has also been noted[29]. Furthermore, kinetic factors are speculated to be more important than thermodynamic factors in determining the crystallographic anisotropy of oxidation[30,31], which is often indicated by the different solute trapping behavior at different crystallographic surface of the alloy[8,30]. Collectively, the crystallographic orientation-dependent oxidation behavior is mostly relied on post-mortem analysis[6,7,28,32-35], which is hard to lead to atomistic understanding for how crystal facets drive unique forms of oxidation resistance[6,33]. Therefore, real-time direct in situ observations[36-38] of the early-stage oxidation processes at the atomic scale on different crystal facets are critically needed to reveal the dynamic mechanisms of crystal facet-depended oxidation and the interplay between kinetic and thermodynamic factors.

In this work, by using in situ environmental transmission electron microscopy (TEM) and density functional theory (DFT) simulation, we directly demonstrate the crystal facet-dependent initial oxidation

[1]Environmental Molecular Sciences Laboratory, Pacific Northwest National Laboratory, Richland, WA, USA. [2]Department of Nuclear Engineering, University of Tennessee, Knoxville, TN, USA. [3]Physical and Computational Sciences Directorate, Pacific Northwest National Laboratory, Richland, WA, USA. [4]Energy and Environment Directorate, Pacific Northwest National Laboratory, Richland, WA, USA. [5]Department of Nuclear Engineering and Radiological Sciences, University of Michigan, Ann Arbor, MI, USA. [6]These authors contributed equally: Shuang Li, Li Yang. ✉e-mail: Daniel.Schreiber@pnnl.gov; gaofeium@umich.edu; chongmin.wang@pnnl.gov

behavior in a model, Cr-lean Ni-5Cr alloy at a moderate temperature of 350 °C. To reveal the crystallographic anisotropy oxidation behavior, we directly observe, at the atomic scale, the oxidation of (111) and (001) facets simultaneously, which unequivocally reveals that the (001) facet, as contrasted with the case of (111) facet, shows superior incipient oxidation resistance. We further demonstrate that the oxidation is kinetically controlled, leading to non-equilibrium phase formation, which shows tendency of evolving toward the thermodynamic equilibrium phase during longer annealing. The atomic origin of the surface facet-dependent initial oxidation resistance is revealed to result from divergent cation (Ni) and anion (O) diffusion dynamics across the various oxide/metal interfaces. These observations establish an interfacial atomic sieving effect that exists between the oxide layer and metal surface, which changes with the exposed metal crystal facet, unveiling why a certain facet, for the initial oxidation, is more oxidation resistance than others. Most dramatically, the present result indicates that crystal facet with a low initial oxidation rate may lead to, counterintuitively, a high-rate steady-state oxidation, providing insight for designing metals with super-oxidation resistance through tailored crystallographic facet orientations.

## Results

### Superior incipient oxidation resistance of (001) *vs* (111)

In situ TEM observations directly demonstrate that the (001) facet exhibits greater resistance to initial oxidation than does the (111) surface. Using a bi-crystal surface geometry expressing both (001) and (111) facets separated by a single twin boundary (TB), it was possible to observe the initial oxidation of both surfaces simultaneously and in real time with atomic resolution. Prior to the in situ oxidation, the specimen was first annealed at elevated temperature to remove the native oxides as detailed in the Methods (Fig. S1). The beginning, pristine surface of the TB and clean (111) and (001) surfaces is presented in Fig. 1a. The atomic progression of oxidation (350 °C, $pO_2$-$1 \times 10^{-4}$ mbar) on both surfaces is depicted by in situ high-resolution transmission electron microscopy (HRTEM) images through the remaining panels of Fig. 1 and in Supplementary Video 1. The frames presented in Fig. 1 are reproduced without any annotations in Fig. S2. On the (111) surface, a continuous oxide layer of one atomic thick is observed by 13 s (Fig. 1b), which continues growing both outward and inward as referenced by the original (111) surface of Ni-5Cr (Fig. 1c). In contrast, the (001) surface is inactive except for a slight surface adsorption. This observation indicates that the (111) surface is the more active surface during initial oxidation while the (001) surface is relatively inactive.

As oxidation continues, the (111) surface oxide grows to intersect the TB forming an additional interface defined by the oxide and the crystallographically equivalent $(\bar{1}\bar{1}1)$ of Ni-5Cr face of the adjacent alloy grain. This transverse oxidation front migrates across the TB, oxidizing the metal underneath the (001) surface through the $(\bar{1}\bar{1}1)$ face rather than the (001) surface, as vividly illustrated by the captured snapshots shown in Fig. 1d–f and Supplementary Video 1. A fast Fourier transformation (FFT) of the HRTEM images from the oxide layers reveals split spots on both the (111) and (001) surfaces (Fig. 1e), indicating that the oxide experiences ~15° lattice rotation (θ) when crossing the TB; this is consistent with the intersection angle between the two exposed surfaces (15.9°). These observations clearly indicate that the interfaces defined by the oxide and metal {111} face allow fast mass transport for continued oxidation, while the interfaces defined by oxide and metal {001} face are far more resistant to initial oxidation.

To confirm the above observation, two additional but well-separated (111) and (001) surfaces were independently oxidized and observed under identical oxidation conditions at 350 °C and $pO_2$-$1 \times 10^{-4}$ mbar as shown in Fig. 2 and corresponding Supplementary Videos 2 and 3. Again the original images of Fig. 2 are reproduced without any annotations in Fig. S3. On the (111) surface, one oxide layer

with rock-salt structure is formed during initial oxidation (Fig. 2a). At 1 s, one atomic layer oxide with surficial O adsorption is observed on the flat (111) surface. After 146 s, the oxide has grown to two atomic layers on the (111) surface. In addition, contrast along diagonal stripes is apparent in the metal near the oxide/metal interface (apparent in HRTEM images at 146 s), indicating a zone of dislocations between the oxide and alloy induced by the misfit strain, which had been reported in the oxidation of Ni-Al alloy previously[7]. These dislocations and stacking faults extend into or are trapped within the growing oxide layer, as apparent in the HRTEM images at 178 s.

Contrasted with the (111) surface, a significant feature of the oxidation process on the (001) surface is that two oxide layers with different morphologies nucleated simultaneously during the initial stage, as shown in Fig. 2b. At 1 s, the pristine (001) surface is atomically flat with no obvious O adsorption. At 136 s and 172 s, the (001) surface without oxidation indicates the incipient oxidation is slower on the (001) surface than (111) surface. While there is a small area with oxide appears on the upper or lower metal surface (refer as {110} surface) at 172 s (marked by the white dash line), which implies the oxidation rate of (110) surface is between the (001) and (111) surface. After 284 s, an oxide island has formed on the (001) surface in addition to a subtle concave step with one atomic layer generated below the original alloy surface. At 400 s, several oxide islands have nucleated, growing via a 3D growth mechanism; these islands subsequently coalesce to form an outer oxide layer. Meanwhile, more concave oxide steps have formed into the original alloy surface, following the layer-by-layer growth mechanism, forming an initial inner oxide layer. As shown in the image at 488 s, the interface between the inner and outer oxide layers coincides with the original alloy surface, indicating both inward (anion-dominated) and outward (cation-dominated) transport processes exist simultaneously[39–41]. It should be noted that the observed dependence of oxidation rate on the crystallographic orientation for the case of Ni-5Cr alloy is similarly true for the case of pure Ni as shown in Fig. S5. Further, it is worth mentioning that the temperature will affect the oxidation behavior where we find the two oxide layers character becomes weak at higher temperature (Fig. S4 at 400 °C).

### Kinetically driven initial oxide formation and evolution

Concomitant with different oxidation rates for the (111) and (001) facets is the formation of different oxide phases and cation distributions, in particular the spatial distribution of the Cr, indicating that faster oxidation leads to the formation of non-equilibrium phases. To reveal this effect, we combine high-angle annular dark field scanning transmission electron microscopy (HAADF-STEM) images and electron energy loss spectroscopy (EELS) analysis to thoroughly resolve the chemical and structural state of oxide layers on both (111) (after 310 s oxidation) and (001) (after 770 s oxidation) surfaces.

The (111) surface features a high density of defects and a higher Cr concentration as compared to the base alloy, as shown in Fig. 3a–c and Fig. S6a. The measured alloy Cr concentration (~5 at.%) is consistent with the nominal alloy and a slight Cr depletion is apparent near the metal/oxide interface. Within the oxide, the Cr concentration increases to ~10 at.% or an enrichment factor of 2 over the nominal alloy. FFTs of HRTEM images show that the oxide is consistent with a rock-salt structure containing some defects (Figs. 2a, 3a, and Fig. S7), which is also consistent with the defected oxide layer observed on the (111) surface in Fig. 1. The high density of crystal defects may result from the high Cr concentration in the oxide. Therefore, we name this oxide layer as a Cr-rich rock-salt (Cr-rich NiO) layer. Based on the HRTEM and HAADF-STEM images of the oxide layer in Figs. 2a, 3a, and Fig. S7, we confirm the orientation relationship (OR) between Cr-rich NiO and Ni-5Cr(111) surface is cube-on-cube, same as in Fig. 1, which is consistent with previous reports[9,42–45].

For the two oxide layers on the (001) surface, EELS analysis (Fig. 3d–f) shows that the inner oxide layer is enriched in Cr (~10 at.%),

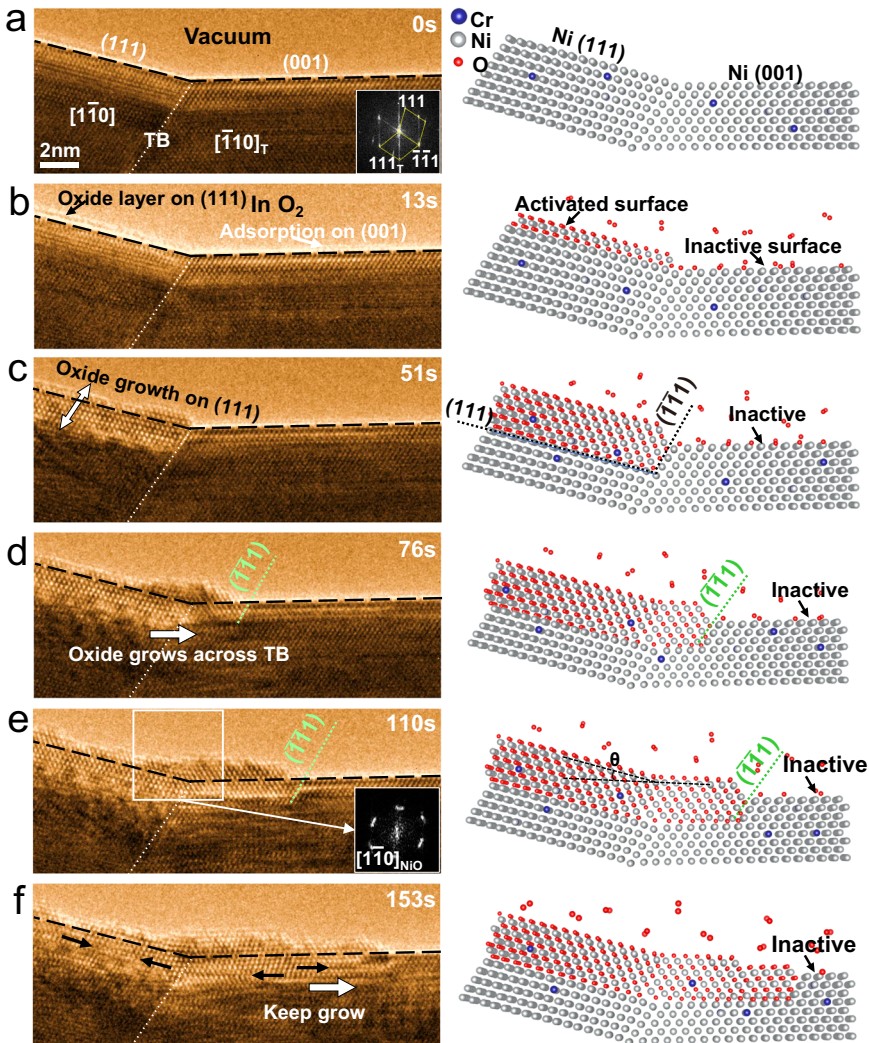

**Fig. 1 | Direct in situ observation of surface facet-dependent oxidation resistance during initial oxidation. a–f** Time-resolved HRTEM images and schematics of atomic oxidation processes on (111) and (001) surfaces beside a twin boundary (TB) in $O_2$ with $p$ - $1 \times 10^{-4}$ mbar and $T = 350$ °C. **a** The pristine clean surfaces under vacuum at 350 °C before in situ oxidation. The insert is the Fast Fourier Transform (FFT) of the image to indicate the crystal orientation beside the twin boundary (TB) as marked by the white dot line in the Ni-Cr alloy. **b** An oxide with one atomic layer nucleated on the (111) surface (indicated by black arrow), while a slight adsorption with bright contrast existed on the (001) surface (indicated by white arrow). **c** The oxide on the (111) surface grows, meanwhile the (001) surface remains inactive.

With respect to the original (111) surface as marked by the black dashed line, the oxide on (111) facet grows both inward and outward as indicated by the white arrows. **d**–**f** As the stoichiometric developed on the (111) facet encounters the TB, an oxide and metal $(\bar{1}\bar{1}1)$ interface is formed, leading to the fast lateral propagation of the oxide layer into the metal section under (001) facet, the oxide, and alloy (111) facet leads the growth of oxide layer as indicated by the green dash line while the (001) surface far from TB is still inactive. The insert FFT in (**e**) display the -15° lattice rotation of oxide crossing the TB. The black arrow in (**f**) indicate the defects (dislocations and stacking faults) in the oxide layer. The black dash lines indicate the original alloy surface position.

while the outer oxide layer has a lower Cr concentration (~3 at.%) and increased Ni concentration (Fig. 3d, e and Fig. S6b.). The Cr concentration in the base alloy shows a similar tendency as the (111) surface, with very minor and localized Cr depletion adjacent to the oxide/metal interface. Combined with the lattice structure analysis of the two layers (Fig. S8), we confirm that the outer oxide layer is a rock-salt (NiO) phase grown with the (OR) of NiO(111)//Ni-5Cr(001) and NiO[$1\bar{2}1$]//Ni-5Cr[$1\bar{1}0$], which, while not the most common OR, has been reported in Ni oxidation previously[46]. For the inner layer, it is harder to confirm a specific oxide phase due to the complex phase and diffraction contrast, and potential overlap of multiple phases. Even so, the Cr-rich inner oxide layer is confirmed to exist in the initial oxidation of (001) surface (Fig. S9). In Fig. S9a, we clearly observe a discontinuous, one atomic layer thick oxide with a Cr-rich concentration as confirmed by the EELS line scan data in Fig. S9c–e. In addition, the geometric phase analysis implies the out-of-plane lattice changes

sharply at the oxide/metal interface (Fig. S9b). Therefore, we term the inner oxide layer as a Cr-rich phase based on the chemical information gained from EELS results.

It is known that Ni oxidation is dominated by Ni cation diffusion, which is orders of magnitude faster than that of O in NiO[47–49]. The diffusivity of Ni is nominally unchanged in the case of Ni-5Cr due to the low Cr concentration[50]. On the other hand, Cr oxidization at a low temperature and low oxygen partial pressure is governed by inward oxygen diffusion[51–53]. Therefore, the element distribution of the outer NiO layer and inner Cr-rich phase layer is consistent with the above discussion that the outer and inner oxide layer are, respectively, limited by outward metal cation (Ni) diffusion and inward oxygen diffusion. Thermodynamically the favored oxidation products of Ni-Cr alloys are rock-salt structured NiO, chromia ($Cr_2O_3$), and spinel ($NiCr_2O_4$), among which $Cr_2O_3$ is most energetically favored[42,53–55]. However, the stoichiometric $Cr_2O_3$ phase is not detected during our in

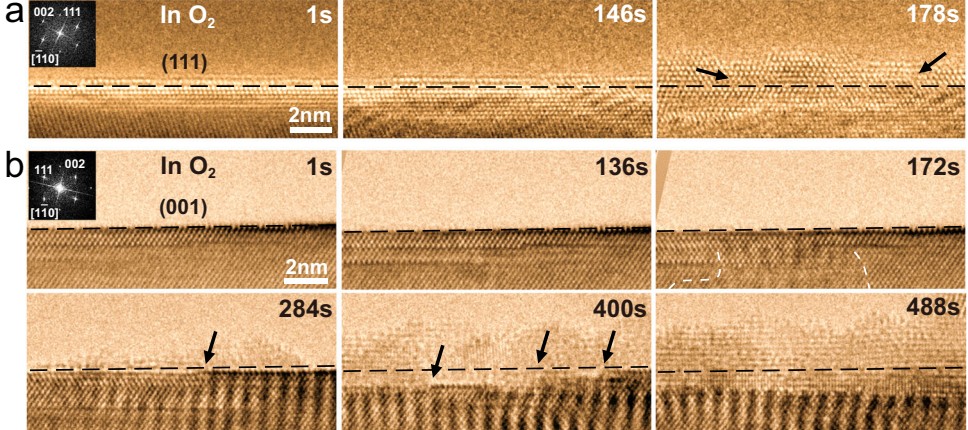

**Fig. 2 | Distinct nucleation and growth process of oxide at (111) and (001) surface of Ni-5Cr alloy during initial oxidation.** Time-resolved HRTEM images reveal the growth of one oxide layer on the (111) alloy surface (**a**) and the two oxide layers on the (001) alloy surface (**b**) in $O_2$ with $p \sim 1 \times 10^{-4}$ mbar and $T = 350\,°C$. The insert is the Fast Fourier Transform (FFT) of the image to indicate the crystal orientation of Ni-5Cr alloy. The black dash lines indicate the original alloy surface. The black arrows in (**a**) indicate the defects (dislocations and stacking faults) in the oxide layer. The black arrows in (**b**) indicate the steps at the alloy surface. The Moiré fringes is due to the oxide layer formed at the upper and lower metal surface (refer as {110} surface) of the thin film.

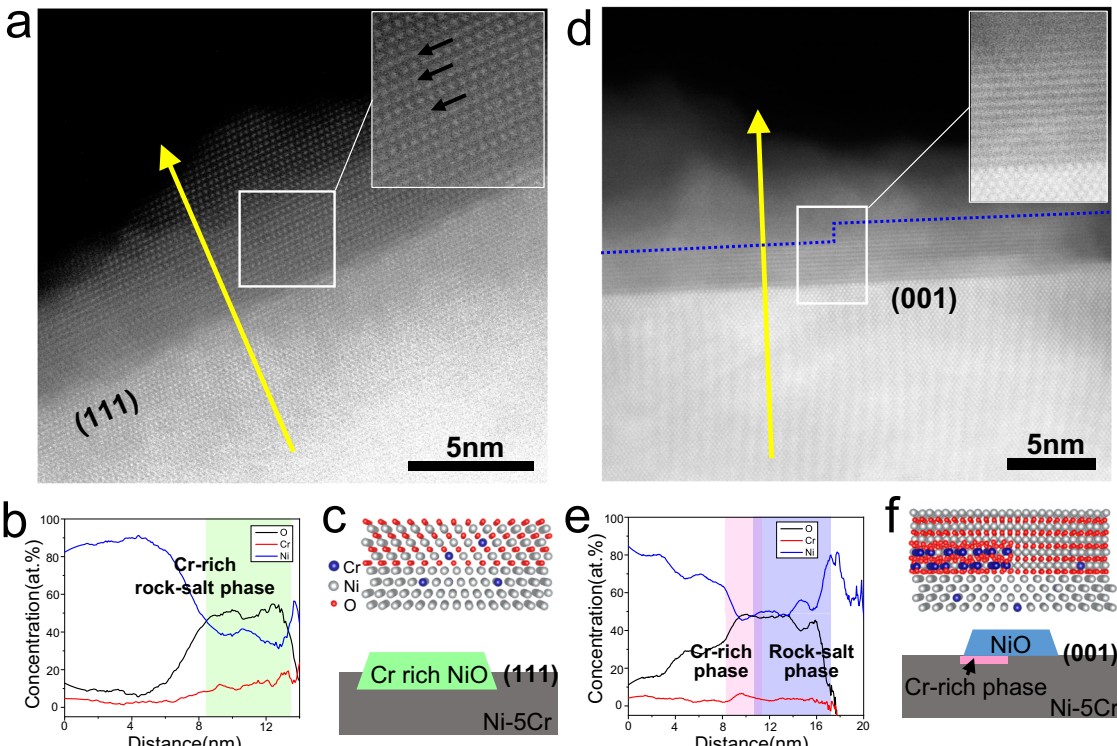

**Fig. 3 | The oxide layers generated at (111) and (001) surface of Ni-Cr alloy during initial oxidation state. a, b** The HAADF-STEM images of the oxide layer (Cr-rich rock-salt phase) with defects (marked by black arrows) at (111) surface at room temperature after in situ oxidization ($O_2$ with $p \sim 1 \times 10^{-4}$ mbar, $T = 350\,°C$) and the relative element concentration line profile obtained from EELS line scan along the yellow line. **d, e** The HAADF-STEM images of the outer (rock-salt phase, i.e., NiO) and inner (Cr-rich phase) oxide layers at (001) surface at room temperature after in situ oxidization ($O_2$ with $p \sim 1 \times 10^{-4}$ mbar, T = 350 °C) and the relative element concentration line profile obtained from EELS line scan along the yellow line. **c, f** The schematics showing the distinct initial oxidation at the (111) and (001) surfaces of Ni-Cr alloy, respectively. The atomic ones indicate the orientation relationship between NiO and alloy.

situ observation of initial oxidation on either (111) or (001) surface, which implies the initial oxidation here is a kinetically limited process. Additionally, the Cr-rich rock-salt oxide layer formed on (111) flat surface had been reported as the consequence of non-equilibrium solute capture in previous studies[8,30,56]. The formation of a non-equilibrium phase further proves the oxidization behavior is kinetically limited,

which can largely be attributed to the relatively low oxidizing temperature and short time duration of these experiments.

**Predicted atomic sieving effect across oxide/metal interface**
The in situ TEM oxidation results clearly indicate that incipient oxidation is faster on the (111) Ni-Cr alloy surface than on the (001)

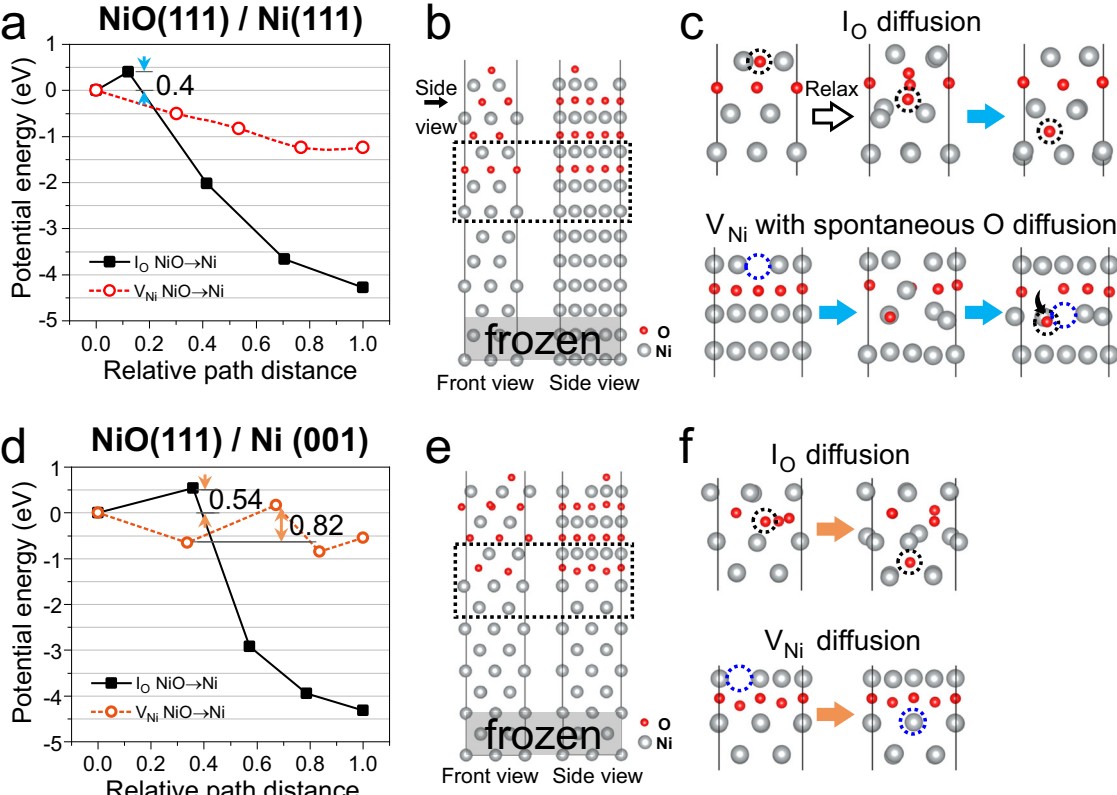

**Fig. 4 | DFT calculation of energy barriers for O and Ni atoms diffusion across the NiO/Ni interface. a**, **d** Activation energy for diffusion profile of an interstitial O atom diffused from NiO to Ni and a Ni vacancy migrated from NiO to Ni (i.e., Ni migrated from Ni to NiO). **b**, **e** The atomic model of NiO(111)/Ni(111) and NiO(111)/Ni(001) interface. The bottom two atomic layers are frozen, which represents the bulk metal. **c**, **f** The enlarged atomic schematics of the black dash rectangle area in (**b**) and (**e**) during the calculation, showing the corresponding diffusion paths of interstitial O and Ni vacancy in (**a**) and (**d**). The diffused O atoms and Ni vacancies are marked by the black and blue dash circles, respectively. The O diffuses across the NiO(111)/Ni(111) interface with double diffusion mechanisms. One is the interstitial diffusion mechanism, which is same as the NiO(111)/Ni(001) interface. The other one is the spontaneous O diffusion along with the Ni vacancy diffusion, referred as cation-vacancy-cooperated (CVC) diffusion mechanism.

surface. Firstly, the surface reaction on different crystal planes was considered by calculated the O adsorption energy as well as the O and Ni diffusion energy on the (111) and (001) surface using density functional theory (DFT). Compared to the (001) surface, the (111) surface has the higher O adsorption energy, lower energy barrier for O jumping into metal, and easier outward Ni diffusion. These calculations indicate that energetically it favors oxide nucleation on (111) surface during the incipient oxidation right before the metal/oxide interface is generated. The detailed discussion and simulation process was presented in Supplementary Section 2.

Besides the surface reaction, the metal/oxide interface can itself act as an additional diffusion barrier, including potential crystallographic differences, leading to a so-called atomic sieving effect across the interface. We believe that these interfacial crystallographic differences can influence oxygen diffusion and explain the resulting oxidation differences. To understand the relevant transport kinetics, it is first important to note that for the NiO layer, Ni cation diffusion is believed to be faster than oxygen anion diffusion. While the case changes to crystallographic dependency when considering the cation and anion diffusion through the interface. In particular, fast diffusion of oxygen through the (111) interface can lead to the rapid formation of NiO that traps Cr within the NiO lattice. In contrast, across the (001) interface, Ni outward diffusion is dominant and Cr becomes trapped primarily at the oxide/metal interface forming a Cr-rich phase with slower oxygen inward diffusion. To substantiate the above interface-governed transport hypothesis, we performed diffusion energy barrier calculations using DFT.

First, the NiO/Ni interface was modeled for Ni (001) and (111) surfaces, and then diffusion energy barriers were calculated for Ni and O across each interface (Fig. 4). These calculations did not include Cr atoms due to the low Cr concentration and limited size of the DFT simulation cells. Further discussion on the role of Cr in diffusion mechanisms at the free surface of NiO and Ni is proposed in Supplementary Section 2.4. Near the NiO(111)/Ni(111) interface, the energy barrier for interstitial oxygen approaching the interface from one Ni layer away in the NiO side is 0.4 eV. Once the interstitial oxygen from the NiO side arrives at the interface, it can cross the interface without an energy barrier and will be strongly trapped in the fcc Ni side (Fig. 4a–c). Notice that the interstitial oxygen diffuses across the NiO(111)/Ni(111) interface spontaneously during the relaxation of the DFT model when the O is within a half atomic layer from the interface, which represents a zero energy barrier. In addition, the near-zero energy barrier for Ni vacancy passing through the interface from the NiO to the fcc Ni side (i.e., Ni atoms migrating from fcc Ni into NiO) indicates a spontaneous process for Ni jumping from metallic Ni to NiO in the presence of a Ni vacancy. Interestingly, O also moves simultaneously from NiO towards the Ni during the Ni vacancy diffusion from NiO to metallic Ni (Fig. 4c), indicating Ni vacancy significantly enhanced spontaneous oxygen diffusion, which is termed as a cation-vacancy-cooperated (CVC) diffusion mechanism. This DFT calculation indicates that oxygen rapidly diffuses from NiO(111) into Ni(111) via two energetically favorable mechanisms: interstitial and CVC mechanisms.

Conversely for the NiO(111)/Ni(001) interface (Fig. 4d–f), the interstitial O and Ni vacancy diffusions both show positive energy barriers, which is consistent with the slower oxide growth observed on

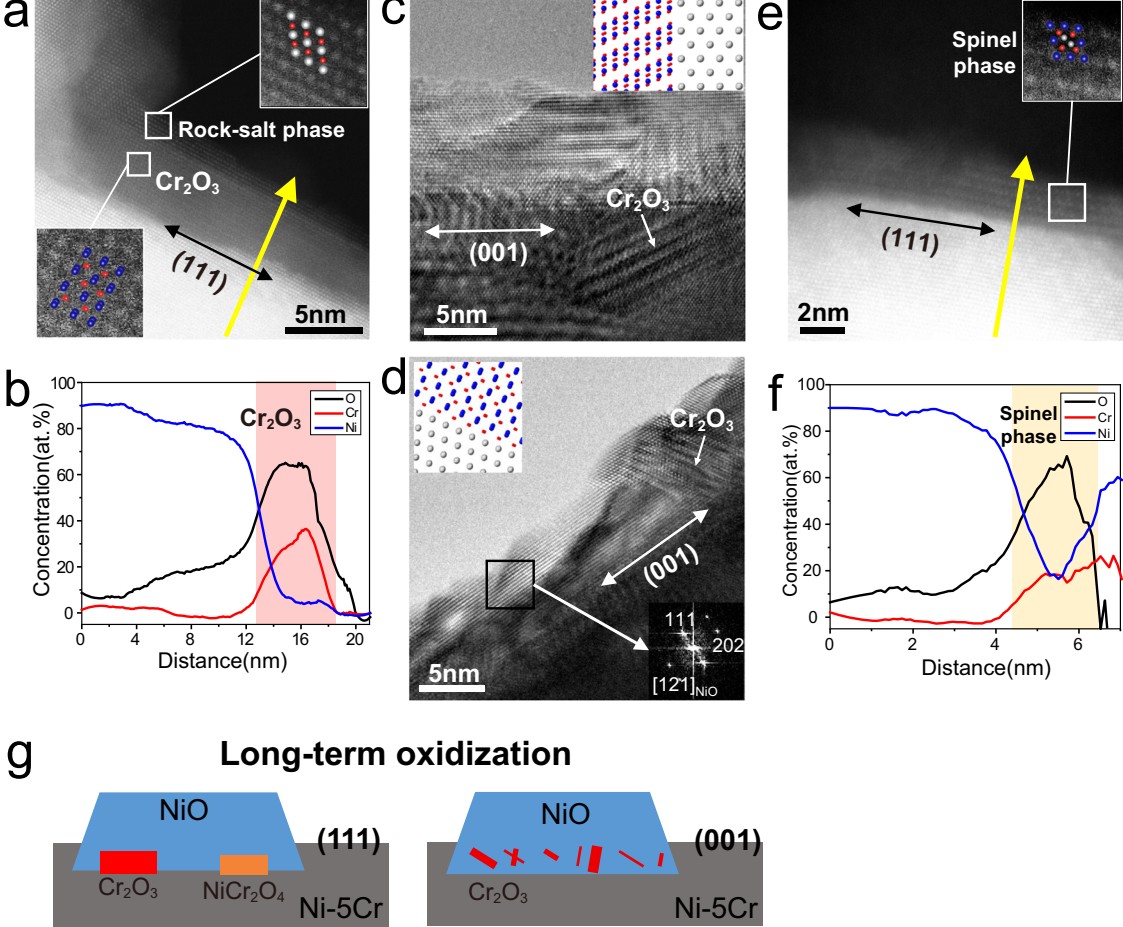

**Fig. 5 | The chromia and spinel phase formation during extended oxidation (more than one hour at 350 °C). a, b** The HAADF-STEM images of the chromia ($Cr_2O_3$) at (111) surface and the relative element concentration line profile obtained from EELS line scan along the yellow line after extended oxidation. **c, d** The HRETM image shows chromia phase generated at (001) surface after extended oxidation. The insert in (**c, d**) is the atomic model shown the orientation relationship between chromia and alloy. **e, f** The HAADF-STEM images of the spinel ($NiCr_2O_4$) phase at (111) surface and the relative element concentration line profile obtained from EELS line scan along the yellow line after extended oxidation. The insert in (**a, e**) is the atomic model overlapping with the enlarged images. **g** The schematic showing the chromia and spinel phase generated at the (111) and (001) surface during extended oxidation.

Ni(001) surface during initial oxidation. Specifically, a comparison of the energy barriers for interstitial O crossing the NiO(111)/Ni(001) and NiO(111)/Ni(111) interfaces (0.54 eV for a position one-half atomic layer away vs 0/0.4 eV for one-half to one atomic layer away) indicates a preferred interstitial O diffusion through NiO(111)/Ni(111) interfaces. In addition, the double O diffusion mechanism at the NiO(111)/Ni(111) interface further promotes the O diffusion rate.

The DFT results reveal that crystallographic interfacial differences between NiO(111)/Ni(111) and NiO(111)/Ni(001) can themselves contribute to a previously unreported atomic sieving effect during oxidation. Based on further mathematical analyses of these calculations, including diffusion through NiO and across the oxide/metal interfaces (described in more detail in Supplementary Section 2.3), the ratio of O diffusion to Ni diffusion is lower for the Ni(001) than the Ni(111) surface. The lower O/Ni diffusion ratio (i.e., slower O diffusion rate) thereby increases the potential flux difference between cations and anions, which consequently results in the outward oxidation direct (cation dominant) and hierarchical oxidation (Cr-rich interface) observed on (001) surface. Conversely, a more balanced cation and anion flux contributes to a solute-trapped and more homogeneous initial oxide on the (111) surface. Combining the in situ experimental and DFT calculation results, we believe the atomic sieving effect across the interface, which highly promotes oxygen diffusion across the (111) surface, is

the dominant factor for the surface-dependent distinct initial oxidation behavior.

## Evolving towards thermodynamic equilibrium phase
Consistent with the fact that the kinetic-driven non-equilibrium phase will experience continued phase evolution, it has been observed that the thermodynamic equilibrium phase was detected upon extended oxidation. Chromia ($Cr_2O_3$) had been observed on both the (111) and (001) surfaces of Ni-5Cr alloy, indicating a continued Cr migration, even the Cr atoms trapped in the NiO for the case of the initial oxidation on the (111) surface.

The HAADF-STEM image in Fig. 5a shows the chromia ($Cr_2O_3$) phase grown at (111) surface and the EELS analysis (Fig. 5b and Fig. S10a, b) indicate $Cr_2O_3$ is nearly the perfect stoichiometry without Ni-doped. The $Cr_2O_3$ phase is also observed at the (001) surface after long-term oxidation (Fig. 5c, d). The orientation relationship (OR) between $Cr_2O_3$ and Ni-Cr is reported previously as $Cr_2O_3$(003) //Ni-5Cr{111}, $Cr_2O_3$ < 210 > //Ni-Cr<110> where the most closely packed {003} planes of $Cr_2O_3$ are semi-coherent with the most closely packed {111} planes of the Ni-Cr alloy[30,54,57]. Considering the fact that $Cr_2O_3$ prefers to grow as rod and/or platelets with the long axis paralleled to an (003) plane during the oxidation of Ni-Cr alloys plane[30,57], the $Cr_2O_3$ could form as platelets lying on the Ni-5Cr(111) surface, while it shows an intersection angle with the Ni-5Cr(001) surface. The former is

consistent with our experimental observation at the (111) surface of Ni-5Cr alloy (Fig. 5a and Fig. S11a), especially in the atomic model with enlarged image. For the latter, the $Cr_2O_3$ lattice orientation indeed shows an intersection angle of either ~26° (Fig. 5c) or 55° (Fig. 5d) with the (001) surface of Ni-5Cr alloy in our experimental results. We build a model based on the OR and find that the intersection angles between the (001) plane of Ni-5Cr alloy and a (012) plane or a (003) plane of $Cr_2O_3$ is 55° and 26° (atomic models as inserts in Fig. 5c, d), respectively, when observed from the <110> zone axis of Ni-5Cr alloy. Therefore, the atomic models match well with the experimental observation. In addition, the rock-salt phase is observed existing outside of chromia phase both at (111) and (001) surface (Fig. 5a–d and Fig. S11), which is unsurprising due to two reasons: (1) $Cr_2O_3$ should be generated near the oxide/metal interface instead of the surface during the following oxidation because of the low diffusion coefficient of Cr; (2) the generated $Cr_2O_3$ is not continuous due to the lower Cr composition of the alloy, which means the oxidation of Ni could not be fully suppressed. The spinel ($NiCr_2O_4$) phase grown at the (111) surface, as shown in Fig. 5e and Fig. S11b, has the cubic-on-cubic OR consistent with previous reports[54,57]. The EELS analysis (Fig. 5f and Fig. S10c, d) confirms the chemical composition of $NiCr_2O_4$, where the deviation of the Ni/Cr ratio may associate with the disorder structure or the overlap of spinel and rock-salt oxides.

According to the low Cr content in the Ni-5Cr alloy and low-temperature heating, there is no long-range outward diffusion of Cr, which means it is hard to generate a continuous and compact $Cr_2O_3$ layer. Therefore, the spinel phase becomes thermodynamically stable and starts to form among the pre-existing non-compact chromia. The fact that $NiCr_2O_4$ is only observed at (111) surface is consistent with the fact that a fast oxidation leads to the $NiCr_2O_4$ phase, while a slow oxidation leads to the stoichiometric $Cr_2O_3$ phase. Even though $Cr_2O_3$ is thermodynamically favored oxide, the $Cr_2O_3$ growth is severely limited by the low Cr concentration and slow Cr diffusion in the alloy during the initial stage. Therefore, our experimental results indicate that the long-term oxidation becomes thermodynamic controlled with the similar oxide products of both $Cr_2O_3$ and NiO oxides, compared to the kinetically dominated initial oxidation.

The steady-state transport kinetics across the oxide/metal interface were further considered computationally using classical DFT (cDFT) within the context of the Poisson-Nernst-Planck (PNP) transport kinetics model (Fig. S16), described further in Supplemental Section 3. During steady-state oxidation, this model predicts that the Cr flux through Ni-5Cr (111) surface is 4.6 times faster than through the (001) surface, driven primarily by differences in chromia formation energies at these two surfaces. In contrast, Ni fluxes are almost identical through the two surfaces, with the (111) being ~1.12 times larger, which reflects similar formation energies of NiO on both surfaces and is qualitatively in agreement with the preceding DFT result for slightly higher Ni flux from the (111) surface. Thus, we conclude that the corresponding oxide formation is driven by several competing factors: the 95-5 ratio of Ni-Cr content in the alloy, over 2 orders of magnitude faster diffusion of Cr compared to Ni, which compensates for the density differences and promotes Cr accumulations at the oxide/metal interfaces, and differences in $Cr_2O_3$ formation energies on (111) and (001) surfaces. Taken together, these factors drive the favorable formation of epitaxial chromia at the (111) surface followed by the formation of NiO further away from the interface with the alloy as illustrated in Fig. 5a and Fig. S11. On the other hand, the significant energy barrier for chromia formation at the (001) surface favors the oxidation of Ni to form NiO, predicting only a single chromia layer not in direct contact with the surface as shown in Fig. 5c, d. It is noteworthy that these simulations reflect the steady-state oxidation conditions and imply the sufficient supply of oxygen to the interfaces, which is most comparable to phase distributions formed after longer oxidation.

It is rather surprising that the observed higher initial oxidation rate on the (111) surface than on the (001) surface appears in partial contradiction to some long-term oxidation studies[16,27,28]. Our in situ observations indicate that, for crystallographically-dependent oxidation, a transition from initial oxidation to longer-term oxidation may be accompanied by a rate inversion. As shown in Fig. S12, further oxidation of the TB region reported in Fig. 1 leads to similar oxide thicknesses of both the (111) and (001) surface, supporting this transition in ongoing oxidation rate. There are likely many contributions to these changes. First, initial oxidation products are oftentimes not representative of more thermodynamically favorable oxides that eventually form during ongoing oxidation. Epitaxial relationships and nucleation rates may also have transient effects. For the case of the Ni-5Cr (111) surface, a dense oxide layer formed upon the initial oxidation, while for the case of (001) surface, a more granular oxide with a higher density of grain boundaries formed at a slightly longer time. Grain boundaries are oftentimes short-circuiting diffusion pathways that facilitate ongoing oxidation, and a greater density of these defects on the (001) surface will certainly impact the ongoing oxidation kinetics beyond the time-scale of these in situ studies[27]. Additionally, the different oxidation environment, such as water, water vapor or the electrolyte contained various ions, will also influence the oxide layer growth, which may explain the previously reported high corrosion resistance of (111) plane in NiCrFe alloy[16]. Furthermore, once the oxide film reaches a certain thickness, the predicted atomic sieving effect will also play a more limited role as mass transport through the thickening oxide, especially along crystalline defects, will become dominant. This point has been similarly observed for the case of passivation of Li and Na for which Li is stable but Na is unstable in dry air[58]. The origin of such a different stability of Li and Na in dry air is rooted in the fact that, upon initial passivation, the $Li_2O$ formed on Li metal is in the form of a compact layer, while the $Na_2O/Na_2O_2$ layers on Na metal are porous and rough.

## Discussion

Direct in situ TEM atomic level observation provides convincing evidence on the crystal surface facet-dependent initial oxidation resistance of alloys, featuring fast oxidation of (111) surface facet as contrasted with the case of (001) for the Ni-5Cr. The fundamental cause of such crystal facet-dependent oxidation rate difference during initial oxidation is rooted on the instantaneous formation of a heterogeneous interface between the metal and the oxide layer. It is apparent that the structure of this interface mimics a 2-dimensional (2D) Moiré layer that is sandwiched between the metal and oxide layer, which correspondingly serves as an atomic sieve and selectively promotes diffusion of certain atomic species. The interfacial modified oxidation is a kinetic controlled process, rather than a thermodynamic controlled process, and correspondingly resulting in the oxide layer and dopant distributions that are in a non-equilibrium phase. In a more general sense, the heterogeneous interface, either pre-existing or instantaneously formed as a consequence of surface response to different environment, can critically modify the mass transport across such an interface. It is apparent that with an increase of the oxide film thickness, the interfacial kinetic controlled surface facet-dependent initial high oxidation rate may reverse to a low oxidation rate at the stage of steady-state oxidation due to the takeover of the prevalent mass transport through the oxide layer film-controlled process. Our findings solve the long-standing mystery of crystal facet-dependent anisotropic initial oxidation behavior. It is further implied that tailoring alloys with certain crystal facet can lead to super-oxidation resistance, which can be reached through oriented growth of thin films or specific surface treatment.

## Methods

### TEM specimen preparation of Ni-5Cr alloy

Ni-5Cr (5.5 at% Cr) binary alloy was prepared in an arc furnace, and then experiences solution anneal heat treatment at 950 °C for 1 h followed by water quenching to eliminates the chemical segregation. The TEM specimens were prepared by using a dual-beam focused ion beam (FEI Helios Dual Beam) microscope with standard lift-out and milling techniques. The final polishing processes were carried out at a low voltage of 2 kV and current of 5.5 pA to minimize the Ga ion beam damage. A Fischione 1040 NanoMill with $Ar^+$ source was used afterward to remove the Ga damage (500 eV, 200 pA). The element composition of the as-prepared TEM specimen is confirmed by the energy-dispersive X-ray spectroscopy (EDS) in Table S1.

### In situ ETEM oxidation experiment

The oxidation experiments were conducted in a FEI Titan ETEM 80–300 equipped with an objective-lens aberration corrector. A Gatan 652 double-tilt TEM heating holder was used to enable TEM specimen temperature control up to 850 °C. The TEM specimen was annealed under high vacuum (~$10^{-7}$ mbar) at elevated temperature (~700 °C) for 30 min to remove the native oxides (Fig. S1a), and then quenched to 350 °C for in situ observation with atomic scale. The cleanliness of the sample was checked by high-resolution TEM and electron diffraction after annealing at 700 and 350 °C, both indicated clean surface without native oxides (Fig. S1b–d). In addition, Fig. S1b–d both indicate the sample tend to have the free surface with low index from the anneal process, which allow us to have the flat (001) and (111) surface for in situ observation. Pure oxygen gas (~99.999%) was introduced into the microscope through a leak valve to oxidize the specimen at 350 °C and $pO_2$-$1 \times 10^{-4}$ mbar. This well-controlled gas delivery system could accurately control the gas pressure from $1 \times 10^{-7}$ to 1 mbar. The temporal resolution of the recording camera is set as 0.2 s for the in situ observation. The electron-beam dose rate was chosen as ~$5 \times 10^4$ e/nm²·s based on the safe electron dose rate for environmental TEM studies with atomic resolution had been established in the published literatures[9,59–61], as discussed in Supplementary Section 1.4.

### Ex-situ microstructure and chemical characterization

The specimen was colling down to room temperature in high vacuum after oxidization for further ex-situ characterization. A 300 kV FEI Titan monochromated (scanning) transmission electron microscope ((S)TEM) equipped with a probe aberration corrector was used to collect ex-situ (S)TEM images and electron energy loss spectroscopy (EELS) to analyses the microstructure and chemical composition of the oxide layers. The EELS collection semi-angle during the spectroscopy experiments was ~45 mrad. EELS spectra dispersion was 0.25 eV/channel with vertical binning at 130 in dual EELS mode.

### DFT simulation

The climbing image nudged elastic band (CI-NEB) method implemented in the Vienna Ab initio Simulation Package (VASP) was used to investigate the energy barriers for various diffusion paths. For the adsorption and diffusion of oxygen near Ni surfaces, as well as Ni diffusion near Ni surface covered with 1 ML O, the generalized gradient approximation (GGA) in the Perdew-Burke-Ernzerhof form was employed for the exchange and correlation interactions. Because of the strong correlation nature of NiO, conventional DFT schemes would predict an artificial narrow bandgap for the ground state. Thus, the GGA + U with an effective Hubbard parameter ($U_{eff}$) proposed by Dudarev et al.[62] has been used to predict the O or Ni diffusion energy barriers near NiO/Ni interfaces. A value of $U_{eff} = 4.3$ eV is chosen according to the effect of $U_{eff}$ on the properties of both fcc Ni and rock-salt NiO. The supercells of surface and interface, as well as some critical parameters in the simulations, are provided in Supplementary Section 2.

### cDFT-PNP simulations

The kinetics of surface oxidation and element profiles were simulated using the mesoscale model based on coupled Poisson-Nernst-Planck (PNP) transport model and classical density functional theory (cDFT), which is used to calculate the free energies and free energy gradients driving atom migration and the formation of the oxides (see SI for details). These include the free energies of Coulomb interactions, electrostatic correlations, hard-sphere repulsion, and short-range interactions with the stationary (lattice) sites, which represent the equilibrium sites for matrix and minor elements in the crystal structure of the alloy and oxide[63,64]. A metal alloy/oxide interface is modeled in 3D as an array of interaction centers representing atomic positions of Cr, Ni, and O in the crystal lattices of the alloy (fcc), NiO, and $Cr_2O_3$ oxide phases. The crystallographic vectors of the Ni fcc lattice are aligned along the Cartesian vectors of the simulation cell. An oxide slab of finite thickness (2 nm) resides at the planar surface of the metal alloy, and the oxide slab is truncated at one end to represent the alloy/oxide interface at the leading edge of the oxidation front. The lattice of interaction centers in the oxide slab corresponds to a superposition of Ni atomic positions in NiO and Cr atoms in $Cr_2O_3$ oxides. The assumption was made that the oxygen concentration in the oxide region is sufficient for forming stoichiometric metal oxides with NiO and $Cr_2O_3$ structures. Simulations were performed at 350 °C matching the temperature employed in the experimental work.

## Data availability

The data generated in this study are provided in the Figures and Supplementary Information. All raw data are stored at the Environmental Molecular Sciences Laboratory of Pacific Northwest National Laboratory and can be provided by the corresponding authors upon request.

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

## Acknowledgements

We thank Libor Kovarik and Bethany E. Matthews for technical assistance. This work was primarily supported by the U.S. Department of Energy (DOE) Office of Science, Basic Energy Sciences (BES), Materials Science and Engineering Division, Mechanical Behavior and Radiation Effects program (PNNL FWP 56909). ETEM measurements were conducted on a project award (DOI: 10.46936/staf.proj.2016.49687/60006123) from the Environmental Molecular Sciences Laboratory, a DOE Office of Science User Facility sponsored by the Biological and Environmental Research program at Pacific Northwest National Laboratory (PNNL). PNNL is a multiprogram national laboratory operated by Battelle for the U.S. DOE under Contract DE-AC05-79RL01830.

## Author contributions

D.S., F.G., and C.W. conceived the idea and designed the experiment. S.L. and J.C. conducted the in situ ETEM and analyzed the data. N.O. carried out the EBSD for FIB lifting out of the sample. L.Y. conducted the DFT simulation with input from F.G. and B.W.. M.S. and P.S. conducted the cDFT-PNP simulation. S.L., L.Y., and C.W. wrote the manuscript with input from all authors and all the authors have given approval to the final version of the manuscript.

## Competing interests

The authors declare no competing interests.
