## [Peer Review File · Nature Communications]

Selective atomic sieving across metal/oxide interface for super-oxidation resistanceREVIEWER COMMENTS

Reviewer #1 (Remarks to the Author):

The authors investigated the initial oxidation behavior of a Ni-Cr alloy and clarified the crystal orientation dependence of oxide layer growth and the interfacial atomic sieving effect through in-situ ETEM and DFT calculations. The observed phenomena, especially in the oxidation behavior between (111) and (100) with a twin boundary, were very interesting, and the novel findings contribute to the progress of material science in corrosion and surface oxidation. However, there are some concerns regarding the content. Please consider the following issues and revise the manuscript before publication.

1. The used sample is a 5% Cr-Ni alloy. The authors discussed the influence of solute trapping and the generation of chromia at the metal/oxide interface based on ETEM. Why did they conduct the experiment on pure Ni before the Ni-Cr alloy? How will the oxidation behavior change without Cr? They should clarify the influence of Cr on crystal-orientation-dependent oxidation behavior using pure Ni.

2. How does the initial oxidation behavior change by altering temperature and oxygen pressure during oxidation? I am very curious about whether the phenomena observed in this study depend on the oxidation reaction rate or not.

3. The authors explained the difference in oxidation rate depending on the crystal plane based on the atomic sieving effect at the metal/oxidation interface. On the other hand, the dissociation energy of the O₂ molecule can be altered by the crystal plane. Discuss such a difference in surface reaction.

4. The previous report on NiCrFe alloy demonstrated that the termination of the (111) plane of the alloy surface enhanced corrosion resistance (Nat. Commun. 2022). As discussed by the authors, the contradiction against the previous studies might be derived from oxidation rate and degree. However, water or water vapor can also affect the oxide layer growth. Discuss this in more detail, focusing on the differences in experimental conditions.

5. In the ball model on oxidation behavior in Figure 1, part of the Cr solute is depicted in the oxide layer. Are there any experimental pieces of evidence that Cr is present in the NiO layer?

Reviewer #2 (Remarks to the Author):

Utilizing in-situ TEM characterization of oxidation alongside DFT calculations, the authors present a novel interfacial atomic sieving effect that restricts the kinetics of initial oxidation, leading to crystal facet dependency. They further demonstrate that, during prolonged oxidation, the relative transport kinetics at the two studied oxide/metal interfaces shift as the thermodynamic equilibrium of the oxide takes effect. The unveiling of this unique sieving mechanism, coupled with an illustration of the transition from initial to steady-state oxidation behavior, are significant findings with potential implications for the development and engineering of oxide-resistant metallic materials.

The reviewer raises concerns primarily about the thin-film effect, noting that oxidation occurs simultaneously on the “upper” and “lower” surfaces of the TEM foils. As illustrated in Figure 2, oxidation at the side surfaces is evident as early as 146 seconds, as indicated by the Moiré fringes. However, the discussion in Fig. S6 appears insufficient to fully address these concerns. Figures 2, S5, and S6 collectively reveal that the Moiré fringes alter direction, particularly near the surface, suggesting interactions between oxides on the “upper” and “lower” surfaces and the interfaces under study. Therefore, the question arises: could the kinetics of initial oxidation be influenced by different lattice mismatches between the oxides formed on the side surfaces (upper and lower) and those on the surface for samples of varying orientations? During extended oxidation, the oxide on the side surfaces might become thicker, potentially dominating the so-called steady-state oxidation. To address this, additional evidence clarifying the impact of side-surface oxidation on the thin film would be valuable in ruling out alternative explanations for the observed phenomena. Some recommendations include:

- Quantifying the oxide thickness on the “upper” and “lower” surfaces of the thin film over time.
- Comparing the oxidation kinetics on the side surface, presumed to be [110], with those on

[111] and [001] surfaces.

- Conducting a more systematic analysis of the Moiré fringes and the orientation relationships of side-surface oxides, particularly examining orientation changes as they approach the surfaces.

A minor suggestion pertains to the introduction, which is somewhat verbose and repetitive over the first three pages, lacking a concise review of current knowledge. Some sentences are overly long and challenging to follow.

Reviewer #3 (Remarks to the Author):

The phenomenon of self-passivation in metallic alloys is a fundamental issue for corrosion and oxidation resistance. In this study, the authors directly demonstrate crystal facet-dependent initial oxidation behavior in a model Ni-Cr alloy, using in situ environmental TEM and DFT calculations. They observe, at the atomic scale, that the (001) facet exhibits incipient oxidation resistance compared to the (111) facet.

One of the notable features in this work is that the atomic origin of surface facet-dependent initial oxidation resistance appears to be attributed to divergent cation and anion diffusion dynamics across oxide/metal interfaces. In contrast to the oxygen diffusion through the (111) surface, diffusion across the (001) surface seems to be slower, trapping chromium primarily at the oxide/metal interface. The DFT calculations in Figure 3 also nicely support these observations, revealing differences in energy barriers for oxygen and nickel diffusion across different interfaces.

We appreciate that the experimentally obtained images and video clips provide a consistent set of evidence, despite the inherent challenge in capturing them. Therefore, we recommend this manuscript for publication and offer several suggestions for improving its quality as follows.

1. We believe that Figure 1 presents a significant series of in situ HREM images in this work. Rearranging all the images on the left-hand column and placing all the schematic

illustrations on the right-hand column would greatly enhance readers' ability to readily discern the distinct behaviors of initial oxidation on the two surfaces.

2. When we took a look at Figure 2, we misunderstood it as if the oxidation on the (001) surface took place much faster, but finally realized that the elapsed times completely differ in the two cases. To avoid such a misunderstanding, we suggest providing two series of images with mutually comparable elapsed times between the (111) and (001) surfaces in Figure 2, if available. Additionally, Figures 2c-2f would be better included in an independent set of figures as Figure 3 for readers. Figures 2c and 2d are too small to recognize.

3. Although thermodynamic equilibrium is claimed to be reached with longer annealing in this manuscript, the appearance of different oxide phases, depending on the surface orientation, in Figure 4 after long-term annealing directly indicates that the system (350C 10^{-4} mbar P_{O_2}) remains kinetically variable. If a system is at thermodynamic equilibrium, the stable oxide phases that exist are determined based on the corresponding equilibrium phase diagram and should, thereby, be identical irrespective of locations in the system.

Response to Reviewer Comments

Reviewer #1 (Remarks to the Author):

The authors investigated the initial oxidation behavior of a Ni-Cr alloy and clarified the crystal orientation dependence of oxide layer growth and the interfacial atomic sieving effect through in-situ ETEM and DFT calculations. The observed phenomena, especially in the oxidation behavior between (111) and (100) with a twin boundary, were very interesting, and the novel findings contribute to the progress of material science in corrosion and surface oxidation. However, there are some concerns regarding the content. Please consider the following issues and revise the manuscript before publication.

Reply: We sincerely thank the reviewer for the very positive comment to our work. As pointed out by the reviewer, the observation of the oxidation behavior between (111) and (100) with a twin boundary is fascinating, conclusively demonstrating the markedly difference on the oxidation behavior between (111) and (100). The specific comments given by the reviewer are very useful. Upon addressing of these points as detailed in the following, the quality of the manuscript is enhanced.

1. The used sample is a 5% Cr-Ni alloy. The authors discussed the influence of solute trapping and the generation of chromia at the metal/oxide interface based on ETEM. Why did they conduct the experiment on pure Ni before the Ni-Cr alloy? How will the oxidation behavior change without Cr? They should clarify the influence of Cr on crystal-orientation-dependent oxidation behavior using pure Ni.

Reply: We thank the reviewer for raising these excellent points. The Ni-5Cr sample is chosen in our work to understand the segregation of different elements, which provides information to understand the effect of Cr on the oxidation behavior specifically of Ni-Cr alloys. This is in the context of a larger research program that seeks to understand how elemental selectivity (i.e., preferential oxidation of Cr over more noble Ni) drives alloy corrosion and ultimately stress corrosion cracking behavior. Therefore, although we agree the behavior of pure Ni is also interesting and makes for a worthy comparison to the alloy, it is tangential to the overarching programmatic aim of understanding alloy behavior..

Indeed, we agree with the reviewer that we need to clarify the influence of Cr on oxidation behavior. Following the reviewer's suggestions, we carried out *in situ* oxidation of Ni at (001) surface as edge and (110) as top/bottom surface in O₂ with $p=1\times 10^{-5}$ mbar and $T=350$ °C as shown in Figure R1. For pure Ni oxidation, as similarly observed for the case of Ni-5Cr alloy, the upper/lower (110) surface of Ni metal is oxidized faster than (001) surface (images at 5.7s and 6s), consistently demonstrating the interfacial atomic sieving effect induced crystal facet dependency of initial oxidation resistance for the oxidation of pure Ni.

To clarify these points, the Figures for the oxidation behavior of pure Ni were added in the supplementary information. We also added the following sentence in the main text. “It should be noted that the observed dependence of oxidation rate on the crystallographic orientation for the case of Ni-5Cr alloy is similarly true for the case of pure Ni as shown in the Supplementary Fig. S5.”

Figure R1. Time-resolved HRTEM images reveal the growth of the oxide layers at the (001) Ni surface in O_2 with $p = 1 \times 10^{-5}$ mbar and $T = 350$ °C. The black dash lines indicate the original alloy surface position. The white dash line marks the oxide on the upper or lower (110) surface. The black arrows indicate the steps at the alloy surface.

2. How does the initial oxidation behavior change by altering temperature and oxygen pressure during oxidation? I am very curious about whether the phenomena observed in this study depend on the oxidation reaction rate or not.

Reply: Thanks to the reviewer for asking this excellent question. We agree with the reviewer that the temperature and oxygen pressure are key factors to influence the oxidation behavior. It would be expected that increased temperature and oxygen pressure will increase the oxidation rate, but the observed dependencies with crystal

surfaces will hold true, so long as the epitaxial relationship of metal and oxide interfaces persists. If, however, the crystallographic orientation relationship between the metal and the oxide is broken due to a high oxidation rate, then what we observed will not be hold true.

To probe this question, we preformed the *in situ* oxidation of (001) surface of Ni-5Cr alloy at 400 °C with $p_{O_2}=1\times 10^{-4}$ mbar as shown in Figure R3. Compared to the case of oxidation at 350 °C as shown in Figure 2b, the oxidation reaction rate at 400 °C is increased. Morphologically, as similar with the oxidation at 350 °C, there are still steps (marked by black arrows) at the alloy surface at 400 °C, which indicates a layer-by-layer growth mechanism. At 230 s and 306 s, the inner and outer oxide layer phenomenon is not that obvious due to the higher temperature. More extreme temperature variations would likely eventually break this consistency, but this suggests that the observed behavior is not highly temperature specific.

We realized that due to the limited pressure range of 10^{-5} mbar to 1 mbar in the column of our ETEM, it is difficult to study the effect of oxygen pressure with a large pressure range. For a large pressure change, a gas flow holder with 1 atm maximum pressure will be the ideal approach.

To clarify this point, we have added the relevant Figures in the supplementary information as Supplementary Fig. S4. At the same time, we add the following sentences in the main text. **“It is worth mentioning that the temperature will affect the oxidation behavior where we find the two oxide layers character becomes weak at higher temperature (Supplementary Fig. S4 at 400 °C).”**

Figure R2. Time-resolved HRTEM images reveal the growth of the oxide layers at the (001) Ni-5Cr alloy surface in O_2 with $p = 1 \times 10^{-4}$ mbar and $T = 400$ °C. The black dash lines indicate the original alloy surface position. The white and yellow dash line marks the oxide on the upper or lower (110) surface. The black arrows indicate the steps at the alloy surface. The yellow arrows indicate the interface between the inner and outer oxide layer.

3. The authors explained the difference in oxidation rate depending on the crystal plane based on the atomic sieving effect at the metal/oxidation interface. On the other hand, the dissociation energy of the O_2 molecule can be altered by the crystal plane. Discuss such a difference in surface reaction.

Reply: We appreciate the very thoughtful suggestions from the reviewer. To answer this question, we calculate the O adsorption energy on the Ni surface as well as O and Ni diffusion energy near the Ni surface, as shown in Figure. R3 (Supplementary Fig. S13).

The most stable adsorption site on the Ni(111) and (001) surfaces for an oxygen atom is referred to as ‘fcc’ and ‘4f’ sites (Figure R3), respectively. The adsorption energies for oxygen on Ni(111) and (001) surfaces are both positive, which means the adsorption process is energetically favorable (Figure R3a). The Ni(111) with higher energy implies that oxygen adsorption is easier on Ni(111) surface, which is consistent with the *in situ* observations of incipient oxidation. As the coverage of oxygen increases, the diminishing adsorption energies implies that further adsorption become sluggish.

In addition to the oxygen adsorption, the first oxide layer nucleation requires either inward diffusion of O into Ni metal or outward diffusion of Ni. The total energy barriers of O atom diffusion from the surface into the second layer of bulk metal are relatively high on both the (111) and (001) surface (Figure R3c), as 4.04 and 3.63 eV, respectively. The average kinetic energy of molecules at above 1000 K will be sufficient to overcome these barriers, which is higher than our experimental temperature. Therefore, the oxide nucleation reactions with oxygen diffusion at our experiment condition are considered as low temperature oxidation controlled by kinetics. Instead of the total energy barrier, the energy barrier of the first jumping event is most important in determining the reaction rate. The oxygen adsorbed on the (111) surface can more easily penetrate into the subsurface than on the (001) surface because of the lower energy barrier of the first jump (2.43eV for (111), 3.17 eV for (001)). In addition, the lower diffusion barrier of the first jump also could be deducted by the straight diffusion path from the ‘fcc’ adsorption site on (111) surface, instead of the step path on (001) surface, which goes across tetrahedral interstitial to octahedral interstitial positions, as indicated by black arrows in Figure R3d. This higher oxygen diffusion barrier of first jump should be one reason for the slower oxide nucleation of (001) surface.

For the surface with fully covered oxygen, the outward Ni diffusion near the (111) surface can occur with an energy of 0.81 eV (Figure R3b). This process is unlikely to be observed for the (001) surface since Ni could not stably reside on the top of the adsorptive oxygen layer. Notice, it does not mean the Ni diffused out will not happen at all on (001) surface because we have not considered the diffusion at the step edges on the not fully covered surface.

To clarify this point, the points discussed above are presented in the Supplementary Information to explain the slower oxide nucleation of (001) surface compared to the (111) surface during the incipient oxidation right before the interface nucleation from the aspect of the surface reaction on different crystal planes.

Meanwhile, we add the brief explanation in the main text as “**Firstly, the surface reaction on different crystal planes was considered by calculated the O adsorption energy as well as the O and Ni diffusion energy on the (111) and (001) surface using density functional theory (DFT). Compared to the (001) surface, the (111) surface has the higher O adsorption energy, lower energy barrier for O jumping into metal, and the easier outward Ni diffusion. These calculations indicate that energetically it favors oxide nucleation on (111) surface during the incipient oxidation right before the metal/oxide interface is generated. The detailed discussion and simulations process was presented in the Supplementary Information.**”

Figure R3 (Supplementary Fig. S13). DFT calculation of the oxygen adsorption energy and diffusion energy barriers for O and Ni atoms on the (111) and (001) surface. (a) Adsorption energy of O atom as a function of oxygen coverage on the surface. (b) Activation energy for diffusion profile of one Ni atom diffused from bulk to the top of adsorption oxygen layer on (111) surface. (c) Activation energy for diffusion profile of one adsorptive O atom diffused from surface into Ni metals. (d) The atomic models show adsorptive O atom diffusion paths (indicated by black arrows) and corresponding energy barriers in (c).

4. The previous report on NiCrFe alloy demonstrated that the termination of the (111) plane of the alloy surface enhanced corrosion resistance (Nat. Commun. 2022). As discussed by the authors, the contradiction against the previous studies might be derived from oxidation rate and degree. However, water or water vapor can also affect the oxide layer growth. Discuss this in more detail, focusing on the differences in experimental conditions.

Reply: We thank the reviewer for insightful and helpful suggestions. We totally agree with the reviewer that what we observed dependence of oxidation rate on the crystallographic orientation may also be changed by other factors, such as oxidation rate, oxidation media such as water or water vapor, and certainly degree of the oxidation as we seen here as comparing the initial oxidation to the steady state oxidation rate. As suggested by the reviewer, we have extended our discussion along this line in the revised version.

To clarify this point, we added the following sentences in the main text with appropriate references cited. “Additionally, the different oxidation environment, such as water, water vapor or the electrolyte contained various ions, will also influence the oxide layer growth, which may explain the previously reported high corrosion resistance of (111) plane in NiCrFe alloy¹⁶.”

5. In the ball model on oxidation behavior in Figure 1, part of the Cr solute is depicted in the oxide layer. Are there any experimental pieces of evidence that Cr is present in the NiO layer?

Reply: We appreciate the great comment from reviewer. Existence of Cr solute in the oxide is indeed experimentally captured based on EELS analysis as shown for the case of the oxide layer on (111) surface during initial oxidation in Figure 3b. The EELS line scan results of the oxide layer formed on (111) surface in Figure 3b (the element concentration line profile) and Supplementary Fig. S6a (the element intensity line profile) indicate the Cr concentration in the NiO layer. Figure R4 presents the EELS spectrum of the oxide layer formed on (111) surface, where the O, Cr, Ni peak are all visible. Therefore, we confirmed the Cr is present in the oxide film. Present observation is also consistent with the solute capture mechanism observed by others. (Q.C. Sherman, P.W. Voorhees, L.D. Marks, “Thermodynamics of solute capture during the oxidation of multicomponent metals”, *Acta Materialia* 181 (2019) 584–594).

To clarify this point, the appropriate EELS results have been added in the Supplementary Information.

Figure R4. The EELS spectrum of the oxide layer formed on (111) surface during initial oxidation.

Reviewer #2 (Remarks to the Author):

Utilizing in-situ TEM characterization of oxidation alongside DFT calculations, the authors present a novel interfacial atomic sieving effect that restricts the kinetics of initial oxidation, leading to crystal facet dependency. They further demonstrate that, during prolonged oxidation, the relative transport kinetics at the two studied oxide/metal interfaces shift as the thermodynamic equilibrium of the oxide takes effect. The unveiling of this unique sieving mechanism, coupled with an illustration of the transition from initial to steady-state oxidation behavior, are significant findings with potential implications for the development and engineering of oxide-resistant metallic materials.

Reply: We sincerely thank the reviewer for the very positive comment to this piece of work. We also thank the reviewer for the constructive suggestions, which help us to enhance the manuscript quality for readability. In the following we address the specific points as suggested by the reviewer.

The reviewer raises concerns primarily about the thin-film effect, noting that oxidation occurs simultaneously on the “upper” and “lower” surfaces of the TEM foils. As illustrated in Figure 2, oxidation at the side surfaces is evident as early as 146 seconds, as indicated by the Moiré fringes. However, the discussion in Fig. S6 appears insufficient to fully address these concerns. Figures 2, S5, and S6 collectively reveal that the Moiré fringes alter direction, particularly near the surface, suggesting interactions between oxides on the “upper” and “lower” surfaces and the interfaces under study. Therefore, the question arises: could the kinetics of initial oxidation be influenced by different lattice mismatches between the oxides formed on the side surfaces (upper and lower) and those on the surface for samples of varying orientations? During extended oxidation, the oxide on the side surfaces might become thicker, potentially dominating the so-called steady-state oxidation. To address this, additional evidence clarifying the impact of side-surface oxidation on the thin film would be valuable in ruling out alternative explanations for the observed phenomena. Some recommendations include:

- Quantifying the oxide thickness on the “upper” and “lower” surfaces of the thin film over time.
- Comparing the oxidation kinetics on the side surface, presumed to be [110], with those on [111] and [001] surfaces.
- Conducting a more systematic analysis of the Moiré fringes and the orientation relationships of side-surface oxides, particularly examining orientation changes as they approach the surfaces.

Reply: We appreciate the reviewer for the insightful comments and suggestions. We agree with the reviewer that the oxidation on the “upper/lower” (110) surface of the thin film is an important phenomenon, which may affect the oxidation effect observed on the side surface of both (111) and (100).

As a first step, as suggested by the reviewer, we have quantified the kinetics of top/bottom surface oxidation. In doing so, we follow the protocol developed by Luo et al for quantifying the top/bottom surface oxidation (S. Wang, Z. Dong, L. Zhang, P. Tsiakaras, P.K. Shen, L. Luo. Atomic Scale Mechanisms of Multimode Oxide Growth on Nickel–Chromium Alloy: Direct in Situ Observation of the Initial Oxide Nucleation and Growth. *ACS Applied Materials & Interfaces*. 2021, 13 (1): 1903-1913).

We measured the oxide generated on each surface in the observed field of view, as shown in Figure R5. The initial Ni-5Cr metal area is used to normalize the increased oxide area ratio of each surface. As expected, the oxidation rate on these three surfaces follows the tendency of (111) > (110) > (001). This result gives us a clue for examining the possible effect of top/bottom surface oxidation on the side surface oxidation.

For the (111) surface, it is easy to understand because the oxidation rate of “upper/lower” (110) surface is believed to be between the (001) and (111) surface. The initial oxidation behavior of (111) surface is happened before the oxide generated on the “upper/lower” (110) surface, as evidenced in the video of (111) surface at 178 s.

For the case of (001) side surface oxidation, while the oxide was generated on “upper/lower” (110) surface before the (001) side surface oxidation in the video of (111) surface at 172s, as shown in Figure R6. In specific, for the (001) surface, we could have this deduction from two aspects. On one hand, the oxide on the “upper/lower” (110) is not rightly on the side surface at 136 s. With progression of oxidation, the oxide on “upper/lower” surface grew to the middle of the (001) side surface at 172 s as marked by the yellow arrow. But the experiment results show the oxide on the side surface grew quickly on the right site (white arrow) beside the interface step. On the other hand, the HRTEM image of oxides on the (001) surface after initial oxidation (Supplementary Fig. S9a) shows the oxides on the right side has the same morphology where the nearly upper/lower surface had not been oxidized.

Based on above observation and discussions, we can generally state that the effect of oxidation of the “upper/lower” (110) surface on the side surface oxidation of (111) and (001) is not significant.

To clarify this point, we added the following sentence in the main text “**While there is a small area with oxide appears on the upper or lower metal surface (refer as {110} surface) at 172 s (as marked by the white dash line), which implies the oxidation rate of (110) surface is between the (001) and (111) surface.**”

Figure R5. The plot of the oxide area ratio (oxide area vs initial metal area) in the observed projection area as a function of time during *in situ* oxidation in O_2 of Figure 2 and supplementary video 2 and 3.

Figure R6. Several more HRTEM images from the supplementary video 3 of the oxidation at on the (001) alloy surface. The white dash lines indicate the oxide on the upper or lower surfaces.

For quantifying the oxide thickness on the “upper” and “lower” surfaces of the thin film over time, we examined the possible approaches in TEM and STEM imaging, we realized that EELS could be a possible approach for measuring the thickness, while we are not sure as how to deconvolute the metal and oxide layer.

As suggested by the reviewer for “Conducting a more systematic analysis of the Moiré fringes and the orientation relationships of side-surface oxides, particularly examining orientation changes as they approach the surfaces.”, we indeed analyzed the Moiré fringes. For the inner region of the sample, the Moiré fringes on either the (111) side surface sample or (001) side surface sample is induced by the slight difference in the interplanar spacing of Ni and NiO generated on the upper/lower (110) surface as expected and demonstrated by the FFT pattern in Figure R7. It would be expected that the overlapping of the oxide on the (111) side surface with oxide on the top/bottom surface of (011) will induce different pattern of Moiré fringes at the edge of the sample. Similar case will be true for the overlapping of the oxide on the (001) side surface with the oxide on the the top/bottom surface of (011). Clearly, as indicated by the FFT pattern from the different regions, there is no obvious difference on the Moiré fringes from the inner region toward the surface region.

As schematically shown in Figure R8, the Moiré fringes spacing can be calculated by the equation: $D = \frac{d_1 d_2}{d_1 - d_2}$.

For the Moiré fringes in Figure R7a and Supplementary Fig. S7, the (11-1) interplanar spacing of NiO and Ni were measured from the FFT as 0.241 nm (d_1) and 0.204 nm (d_2), while the Moiré fringes spacing was measured as 1.35 nm (D), which is consisting with the calculations.

For the Moiré fringes in Figure R7b and Supplementary Fig. S8, the (220) interplanar spacing of NiO and Ni were measured from the FFT as 0.124 nm (d_1) and 0.146 nm (d_2), while the Moiré fringes spacing was measured as 0.83 nm (D), which is also consisting with the calculation.

To clarify these points, we have added these analyses and discussions as detailed above with corresponding Figures in the Supplementary Materials.

Figure R7. Several more HRTEM images from the supplementary video 3 of the oxidation at on the (001) alloy surface. The white dash lines indicate the oxide on the upper or lower surfaces.

Figure R8. The schematic diagram of the Moiré fringes generated due to the up and down phase with slightly different interplanar spacing.

A minor suggestion pertains to the introduction, which is somewhat verbose and repetitive over the first three pages, lacking a concise review of current knowledge. Some sentences are overly long and challenging to follow.

Reply: We thank the reviewer for the critical suggestions. Upon revision, the introduction has been carefully revised to clarify the points as the reviewer has mentioned. The changes are highlighted.

Reviewer #3 (Remarks to the Author):

The phenomenon of self-passivation in metallic alloys is a fundamental issue for corrosion and oxidation resistance. In this study, the authors directly demonstrate crystal facet-dependent initial oxidation behavior in a model Ni-Cr alloy, using in situ environmental TEM and DFT calculations. They observe, at the atomic scale, that the (001) facet exhibits incipient oxidation resistance compared to the (111) facet.

One of the notable features in this work is that the atomic origin of surface facet-dependent initial oxidation resistance appears to be attributed to divergent cation and anion diffusion dynamics across oxide/metal interfaces. In contrast to the oxygen diffusion through the (111) surface, diffusion across the (001) surface seems to be slower, trapping chromium primarily at the oxide/metal interface. The DFT calculations in Figure 3 also nicely support these observations, revealing differences in energy barriers for oxygen and nickel diffusion across different interfaces.

We appreciate that the experimentally obtained images and video clips provide a consistent set of evidence, despite the inherent challenge in capturing them. Therefore, we recommend this manuscript for publication and offer several suggestions for improving its quality as follows.

Reply: We sincerely thank the reviewer for evaluating our work and the very positive comment to this piece of work. Upon revision, as detailed in the following, we have carefully addressed the comments and suggestion as given by the reviewer, which correspondingly enhance the quality of the manuscript.

1. We believe that Figure 1 presents a significant series of in situ HREM images in this work. Rearranging all the images on the left-hand column and placing all the schematic illustrations on the right-hand column would greatly enhance readers' ability to readily discern the distinct behaviors of initial oxidation on the two surfaces.

Reply: We appreciate the reviewer for this insightful and excellent suggestion. We totally agree with the reviewer.

To clarify this point, upon revision, we rearrange the panels in Figure 1 as suggested, indeed making it easier for the readers to discern the information directly from the Figure.

2. When we took a look at Figure 2, we misunderstood it as if the oxidation on the (001) surface took place much faster, but finally realized that the elapsed times completely differ in the two cases. To avoid such a misunderstanding, we suggest providing two series of images with mutually comparable elapsed times between the (111) and (001) surfaces in Figure 2, if available. Additionally, Figures 2c-2f would be better included

in an independent set of figures as Figure 3 for readers. Figures 2c and 2d are too small to recognize.

Reply: We thank the reviewer for the thoughtful suggestions. We totally agree with the reviewer about this point.

To clarify this point, upon revision, we revised the Figure 2 as the oxidation on (001) and (111) surface with the similar elapsed times with three more images of (001) surface due to its high oxide resistance. We also make the corresponding HAADF images and EELS data as Figure 3 with better recognition. The Figure legends and the main text were also appropriately modified.

3. Although thermodynamic equilibrium is claimed to be reached with longer annealing in this manuscript, the appearance of different oxide phases, depending on the surface orientation, in Figure 4 after long-term annealing directly indicates that the system (350°C 10^{-4}mbar $p\text{O}_2$) remains kinetically variable. If a system is at thermodynamic equilibrium, the stable oxide phases that exist are determined based on the corresponding equilibrium phase diagram and should, thereby, be identical irrespective of locations in the system.

Reply: Thanks to the reviewer for raising the key point, pointing out possible ambiguity the readers may have. We totally agree with the reviewer about this point. Indeed, initially, the formation of different phases and their spatial distribution are kinetically controlled. While with longer time annealing under the oxidation condition of $p\text{O}_2=1\times 10^{-4}\text{mbar}$ and 350°C does not necessarily bring the system to the state of thermodynamic equilibrium. At this sense, the evolution of kinetically dominated state to thermodynamic dominated state is implied by the generation of thermodynamic equilibrium phase.

To clarified this point, we modify the main text reading as: “We further demonstrate that the oxidation is kinetically controlled, leading to non-equilibrium phase formation, which shows tendency of evolving toward thermodynamic equilibrium phase during longer annealing.”

REVIEWERS' COMMENTS

Reviewer #1 (Remarks to the Author):

The authors responded accurately and sincerely to the reviewers' comments. The additional data obtained are critical to the understanding of the paper's content and strengthen the main findings of the study. Therefore, the reviewers strongly recommend that this paper be published in Nature Communications.

Reviewer #2 (Remarks to the Author):

The authors have sufficiently addressed my concerns and comments.

Reviewer #3 (Remarks to the Author):

The authors have addressed my previous comments and concerns and revised their manuscript accordingly. I believe the manuscript is now suitable for publication and thus recommend its rapid publication.

Response to Reviewer Comments

Reviewer #1 (Remarks to the Author):

The authors responded accurately and sincerely to the reviewers' comments. The additional data obtained are critical to the understanding of the paper's content and strengthen the main findings of the study. Therefore, the reviewers strongly recommend that this paper be published in Nature Communications.

Response:

Thanks to the reviewer for the positive comments.

Reviewer #2 (Remarks to the Author):

The authors have sufficiently addressed my concerns and comments.

Response:

Thanks to the reviewer for the positive comments.

Reviewer #3 (Remarks to the Author):

The authors have addressed my previous comments and concerns and revised their manuscript accordingly. I believe the manuscript is now suitable for publication and thus recommend its rapid publication.

Response:

Thanks to the reviewer for the positive comments.